# Effects of Fire Location and Forced Air Volume on Fire Development for Single-Ended Tunnel Fire with Forced Ventilation

**Jinlong Zhao [1], Zhenhua Wang [1], Zhenqi Hu [1], Xinyuan Cui [1], Xiandu Peng [2,\*] and Jianping Zhang [3]**

1    School of Emergency Management and Safety Engineering, China University of Mining and Technology-Beijing, Beijing 100084, China
2    University of Chinese Academy of Social Sciences, Beijing 102488, China
3    FireSERT, Belfast School of Architecture and the Built Environment, Ulster University, Newtownabbey BT37 0QB, UK
\*    Correspondence: zqt2210102036@student.cumtb.edu.cn

**Abstract:** Single-ended tunnels are a typical structure and an important part during tunnel construction. In the case of a fire in a single-ended tunnel, forced ventilation is commonly used to create a safe area near the excavation face. This work is aimed at examining the effects of fire location and air volume on fire development for single-ended tunnel fires with forced ventilation. A single-ended tunnel was built in Fire Dynamics Simulator (FDS), and twenty simulation tests were carried out. In the simulation, the distribution of flow field, temperature, and CO concentration in the tunnel were measured and analyzed. The results show that three regions can be identified based on airflow directions and velocity: (1) turbulent flow zone, (2) turbulent flow transition zone, and (3) steady flow zone. It was found that the maximum ceiling temperature rise decreases first with the distance between the fire source and the excavation face ($X_L$), and then increases with a further increase in $X_L$. The simulation results also showed that CO can easily accumulate on the ventilation duct side at the fire source position and the opposite side of the ventilation duct 5.0–15.0 m downstream of the fire source. Both the CO concentration and the maximum ceiling temperature rise decrease with increasing air volume, while the larger forced air volume will result in a higher risk for the downstream regions. The present results are of practical importance in firefighting and personnel evacuation in single-ended tunnels with a forced ventilation system.

**Keywords:** single-ended tunnel; FDS simulation; fire location; forced air volume; temperature distribution; CO concentration

## 1. Introduction

Single-ended tunnels are a common and important structure during tunnel construction. Compared with a running tunnel, the front end of a single-ended tunnel is sealed, and the space is more limited. In order to control the working environment conditions of the excavation face, forced ventilation by blowers is commonly used [1]. During tunneling (tunnels excavated using the mining method), combustible materials such as belts, cables, wood, etc., are usually present. These storage materials are highly prone to form tunnel fires in the case of ignition, usually resulting in disastrous consequences. For example, a tunnel fire accident occurred at Liangbaosi in 2019, during which 11 people were trapped in the excavation face with the fire located 200 m away from the excavation face, as schematically shown in Figure 1 [2]. During the accident, the forced ventilation air volume was increased by the rescue experts to prevent the smoke from moving towards the excavation face. Eventually, the 11 trapped people were successfully rescued. This accident highlights the great significance of studying the effects of forced ventilation air volume on fire development in single-ended tunnels.

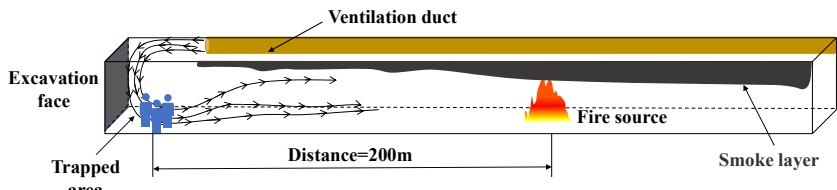

**Figure 1.** Schematic diagram of the fire accident at Liangbaosi.

In recent years, fire and smoke behavior in sealed (the exit is blocked to some extent) or single-ended tunnels has been studied by some scholars, especially for smoke temperature and CO concentration. Wang et al. [3] conducted a series of simulation tests using a small-scale model tunnel (1:10) to investigate the ceiling temperature with different blockages in a tunnel. They found that the vertical temperature and temperature in the downstream direction decreased as the width–blockage ratio decreased. Chen et al. [4,5] conducted experiments with multiple fires to study the sealing effects (sealing ratios: 0, 25, 50, 75, and 100%). A critical sealing ratio was found, at which the ceiling temperature inside the tunnel reached the maximum. Shi et al. [6] investigated the effect of different sealing ratios (0, 65, 75, 85, and 100%) on the self-extinction of fires by propane with different heat release rates (2.8 kW–11.2 kW) in a reduced-scale model (1/20) tunnel. The results showed that most of the self-extinction occurred at a sealing ratio of 75%. Han et al. [7] performed reduced-scale (1/15) experiments to study the blockage effect. It was found that the ceiling temperature decreased as the fire-blockage distance increased, and an empirical model was subsequently established to correlate the decay of the ceiling temperature upstream of the fire with the fire-blockage distance. Han et al. [8] investigated the distribution of the downstream temperature and smoke movement in a sloping tunnel with a one-ended portal based on theoretical analysis and model-scale experiments. The results showed that the fire plume tended to incline towards the excavation face in the tunnel with or without slope when the fire was located near the end wall. Li et al. [9] numerically studied the longitudinal distribution of temperature rise and CO concentration in a single-ended tunnel without ventilation. The results showed that the temperature rise decays as an exponential function, and the CO concentration decays as a power function, while the difference between them increases as an exponential function. These studies showed the fire development in tunnels (smoke movement and CO concentration distribution) is closely related to the sealing of the tunnel. The ceiling temperature can reach the maximum in the critical sealing ratio, and then decrease as the sealing ratio increases.

Apart from the sealing effect on tunnel fire development, in practical accident cases, fire location and ventilation can affect fire development in tunnels. Yao et al. [10] conducted a series of experiments in a 1:5 small-scale model tunnel (L: 14 m, W: 0.4 m, H: 0.5 m), focused on the smoke movement and the temperature distribution beneath the ceiling in an enclosed tunnel. They found that fire location has a significant influence on the maximum smoke temperature beneath the ceiling. Xiao et al. [11] investigated longitudinal ventilation systems in a tunnel fire using a computational fluid dynamics (CFD) model. It was reported that at the initial stage of the fire, the velocity of ventilation fans can be set up at a critical speed to slow down smoke movement and high-temperature diffusion. Tao et al. [12] established a model tunnel to investigate the impact of longitudinal velocity, bifurcated shaft exhaust velocity, and fire location on ceiling temperature and decay. The results showed that fire source location mainly affects the magnitude of the temperature attenuation factor. Kong et al. [13] conducted a series of simulations to investigate the effect of longitudinal fire locations on smoke behavior characteristics induced by inclined tunnel fires. Theoretical analyses and numerical simulations indicated that the influence of fire source location variation on back-layering length depends on downstream length, rather than upstream length. The above studies clearly indicated that fire location and ventilation can significantly affect tunnel fire development. However, limited research has been conducted to study the influence of forced ventilation on fire development in

single-ended tunnels, which is crucial to ensuring the survival of trapped people in the case of fire.

To fill this knowledge gap, this study is aimed at investigating the effects of air volume and fire location on flow field, temperature, and CO concentration distribution in a single-ended tunnel. A series of simulations were carried out using the CFD model, Fire Dynamics Simulator (FDS, version 6.7.9). Important design parameters were analyzed in detail, and the thermal hazard at the excavation face was also studied.

## 2. Numerical Model and Model Validation

### 2.1. Model Construction

The high cost of full-scale experiments seriously hampers its practical usages [14]. Under this circumstance, reduced-scale experiments have been widely used based on the Froude scaling law, as shown in Table 1 [15,16].

**Table 1.** A list of Froude scaling correlations.

| Type of Unit | Scaling |
|---|---|
| Heat release rate (HRR)/(kW) | $Q_F/Q_M = (L_F/L_M)^{5/2}$ |
| Time/(s) | $t_F/t_M = (L_F/L_M)^{1/2}$ |
| Temperature/(K) | $T_F = T_M$ |

Where $Q$ is the heat release rate (kW), $L$ is the tunnel length (m), and the subscripts $M$ and $F$ represent the model tunnel and the full-size tunnel, respectively. Fire Dynamics Simulator is a popular computational fluid dynamic (CFD) code widely used to solve fire-driven fluid-flow-related problems in the fire simulation area. It was compiled by the National Institute of Standards and Technology (NIST). Considering typical underground single-ended tunnels, the FDS model is 100 m (length) by 3.2 m (width) by 4.8 m (height). These parameters are also consistent with those used by Han et al. [17] in a reduced-scale tunnel (1:10). A circular ventilation duct with a diameter of 1.0 m was used for forced ventilation, and the ventilation duct was located near the back wall and ceiling of the tunnel, as shown in Figure 2. According to the requirements of Safety Regulations in Coal Mines [18], the opening of the ventilation duct was 5.0 m away from the excavation face. In the simulations, a series of thermocouples were placed 0.2 m underneath the tunnel ceiling to record the longitudinal temperature distribution. The thermocouple intervals near the excavation face were 1.5 m (a total of 30) and 3.0 m for the regions further away (a total of 19). A total of 49 thermocouples were employed, and the detailed setup is shown in Figure 2.

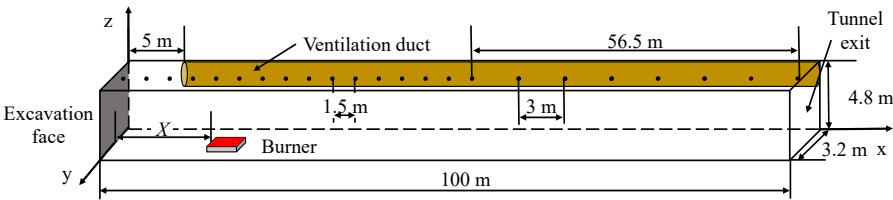

**Figure 2.** Schematic diagram of FDS model.

### 2.2. Fire Parameters and Boundary Conditions Setup

In order to ensure the regular work of the excavation faces, combustible materials such as belts, cables, wood, etc., are usually prepared or stacked in single-ended tunnels. In the simulation, the heat release rate of the fire source was set as 2 MW. The design fire curve was assumed to be a t-squared fire (~$t^2$), which implies that the heat release rate becomes constant after reaching the maximum [19]. The analysis in this paper comes from the stable stage. To reach the stable stage faster, $\alpha = 200$ was selected. The combustion heat in the simulation was set to 48,066 KJ /Kg, and the Mixture Fraction Combustion Model

was selected as the combustion model. The CO yield of the fire-resistant belt was set as 0.06. This value is from our previous cone calorimeter tests, in which the radiant heat is 20 kW/m$^2$. In the simulation, the fixed wall surfaces of the full-size tunnel are treated as thermally insulated surfaces. The environmental temperature in the tunnel was 20 °C.

### 2.3. Meshes

In FDS simulations, the mesh size plays a very important role in improving the accuracy of simulation results and saving computer resources [3]. A smaller grid size can provide more information and better accuracy. However, it needs more computing resources and longer computing time. Therefore, the optimal grid size should not only guarantee the accuracy of simulation results, but also save the computational cost as much as possible [20]. In FDS, the $D^*/\delta_x$ criterion has been widely used to evaluate the resolution of the grid, where $\delta_x$ is the grid size, and $D^*$ is the characteristic diameter of the fire, which is calculated as:

$$D^* = \left(\frac{\dot{Q}}{\rho_\infty c_p T_\infty \sqrt{g}}\right)^{\frac{2}{5}} \tag{1}$$

where $\dot{Q}$ is the heat release rate (kW), $\rho_\infty$ is the ambient air density (1.2 kg/m$^3$), $c_p$ is the specific heat capacity of air at constant pressure (1.0 kJ/kg K), $T_\infty$ is the ambient air temperature (293 K), and $g$ is the gravitational acceleration (9.81 m/s$^2$). It was recommended that the value of $D^*/\delta_x$ should be in the range of 4–16 [21]. Based on Equation (1) and the heat release rate in the simulation, the grid size range can be calculated as shown in Table 2. In all the simulations, a grid size of 0.2 m was used.

**Table 2.** The grid size range with HRR.

| HRR/(kW) | $D^*$/(m) | Grid Size Range/(m) |
|---|---|---|
| 2000 | 1.269 | 0.079–0.317 |

### 2.4. FDS Model Verification

The accuracy of the simulation results was firstly validated against the experimental data obtained in small-scale experiments conducted by Han et al. [17]. In their experiments, the heat release rate was 15.65 kW. For a full-scale simulation, the corresponding heat release rate is 4.95 MW based on the scaling law in Table 1. The smoke temperature rises in the longitudinal direction can be used to explain fire development. Figure 3 shows a comparison of the experimental and predicted ceiling temperature rise under windless conditions.

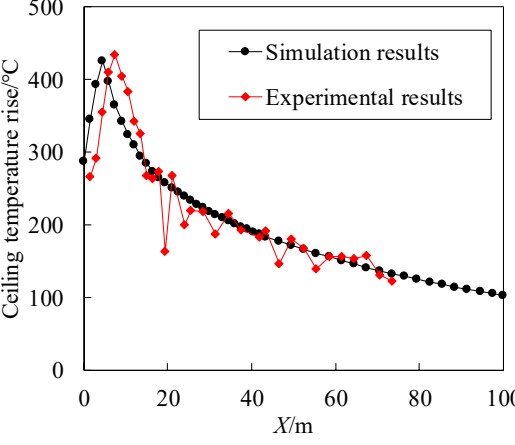

**Figure 3.** Comparison of ceiling temperature rise by simulation results and experimental results.

The simulation results have a good agreement with the experimental data, especially away from the excavation face. The deviation of temperatures is less than 10%, which shows that the setting of the simulation is consistent with the actual situation. There is some difference between the maximum temperature near the excavation face. The reason is twofold. Firstly, the space at the excavation face is limited and the flow field is complex, while the flow field in the experiment is different from the simulation. Secondly, the tunnel sealing is different between the experiments and the simulations. Furthermore, it can be seen in Figure 3 that the experimental temperature suddenly drops at $X = 20$ m, which has a great error with the simulation results. This could be attributed to the poor sealing because the exhaust outlet was set at the top of the experimental tunnel at $X = 20$ m.

### 2.5. Simulation Tests

The distance between the fire source and the excavation face ($X_L$) and the air volume of the ventilation duct ($Q_{air}$) were selected as independent variables. The fire source areas in the simulation were set to 4 m$^2$ (2.0 m $\times$ 2.0 m). For comparison purposes, a simulation without fire was also conducted. Detailed information about the simulation conditions is given in Table 3.

**Table 3.** Simulation Tests.

| Test No. | HRR/(kW) | $X_L$/(m) | $Q_{air}$/(m$^3$/min) |
|---|---|---|---|
| Model Validation Group | 4948.97 | 5.0 | 0 |
| 1 | | | 400 |
| 2 | No fire control group | | 800 |
| 3 | | | 1200 |
| 4 | | | 1600 |
| 5 | | | 400 |
| 6 | | 2.0 | 800 |
| 7 | | | 1200 |
| 8 | | | 1600 |
| 9 | | | 400 |
| 10 | | 5.0 | 800 |
| 11 | | | 1200 |
| 12 | | | 1600 |
| 13 | 2000 | | 400 |
| 14 | | 20.0 | 800 |
| 15 | | | 1200 |
| 16 | | | 1600 |
| 17 | | | 400 |
| 18 | | 50.0 | 800 |
| 19 | | | 1200 |
| 20 | | | 1600 |

## 3. Results and Discussion

### 3.1. Flow Field with Forced Ventilation

To better understand the flow field characteristics of forced ventilation, the cases of $Q_{air} = 1600$ m$^3$/min are taken as an example. The airflow directions of the X–Z section and the X–Y section at a height of 4.3 m in the tunnel under forced ventilation conditions are shown in Figure 4. As shown in Figure 4, the fresh air from the ventilation duct mouth flows towards the excavation face, and after impinging, the airflow direction reverses. The airflow in this region is chaotic, and pockets of fresh air accumulate here due to the restriction by the excavation face and the tunnel walls, resulting in different air velocities in

the tunnel. In practical conditions, this process can cause harmful gases to flow out of this area and guarantee worker safety.

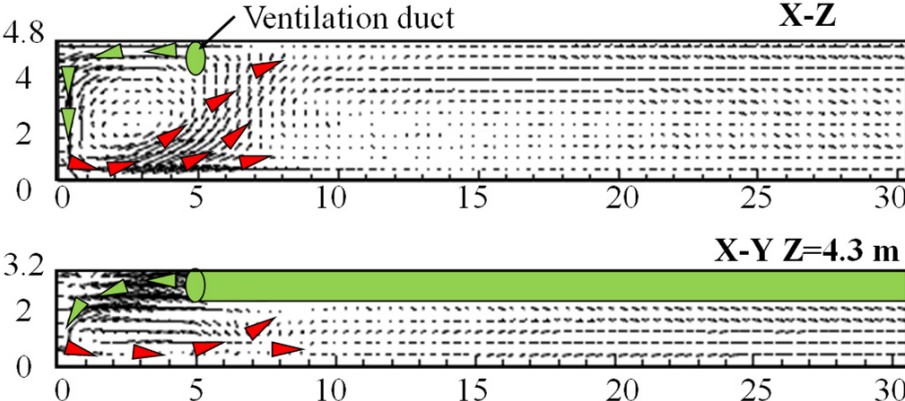

**Figure 4.** Sketch of the airflow movement directions in tunnel near excavation face.

The overall longitudinal flow field of the tunnel from the average over the steady period is shown in Figure 5. The flow field can be divided into three zones based on the airflow velocity and direction: (1) turbulent flow zone, (2) turbulent flow transition zone, and (3) steady flow zone. For the turbulent flow zone, the airflow is chaotic, and a vortex zone is formed between the ventilation duct and excavation face [22], because the airflow is restricted by the excavation face and the side wall [23]. The airflow direction changes after impinging on the wall, and the velocity is high at the bottom of the tunnel (around 5 m to the excavation face). Subsequently, the airflow direction becomes nearly the same, and the airflow velocity decreases with the distance from the excavation face. In the event of a fire in this zone, the vortex can easily lead to backflow, resulting in smoke accumulation near the excavation face. For the turbulent flow transition zone (95% of the airflow direction remains the same), the airflow direction remains relatively stable in the longitudinal direction, and the airflow velocity gradually becomes uniform. The airflow velocity is high near the ceiling and decreases along the vertically downward direction. In the event of a fire in this zone, the airflow near the ceiling would move faster, which could accelerate the smoke spread. For the steady flow zone (95% of the airflow velocity remains the same), both the airflow velocity and directions remain stable. In the event of a fire in this zone, the smoke layer remains steady. The maximum airflow velocity ($V_{max}$) decreases as the air volume decreases. For example, it decreases from 21.065 to 5.476 m/s as the air volume decreases from 1600 to 400 m$^3$/min.

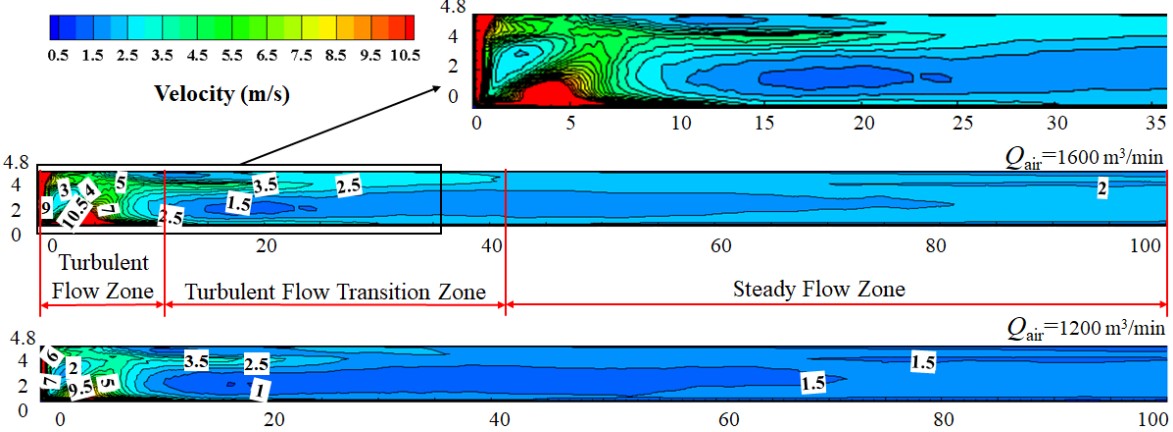

**Figure 5.** Longitudinal flow field diagram of single-ended tunnel.

Air volume also affects the lengths of the three zones (turbulent flow zone, turbulent flow transition zone, and steady flow zone), as shown in Figure 6. The length of the turbulent flow zone remains constant as the air volume increases, because the turbulent flow zone is mainly affected by the vortex formed between the ventilation duct and the excavation face (as shown in Figure 5), and the vortex is determined by the position of the ventilation duct mouth, independent of the air volume. As the air volume increases, the length of the turbulent flow transition zone gradually decreases, while the length of the stable zone shows an opposite trend. The airflow in the turbulent flow transition zone is affected by the vortex to a certain extent [24]. The intensity of the vortex increases with increasing air volume, resulting in airflow velocity reaching uniformity faster.

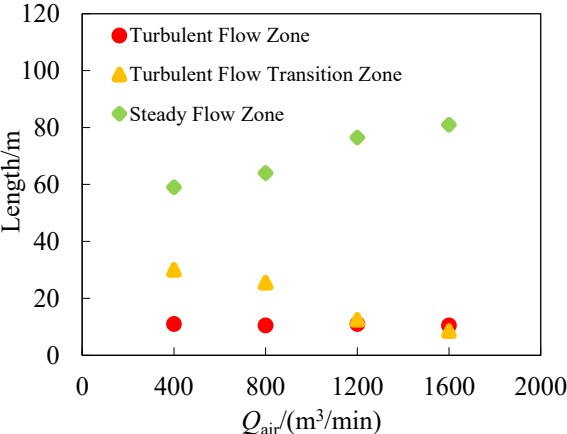

**Figure 6.** The lengths of the three zones under different forced air volumes.

### 3.2. Fire Location Effect on the Internal Environment of Tunnel

#### 3.2.1. Fire Location Effect on the Flow Field Distribution

The distribution of the flow field in the tunnel with different fire locations from the average over the steady period is shown in Figure 7 for the case of $Q_{air}$ = 1600 m³/min. The flow field at the excavation face and downstream of the fire source nearly remains the same. The maximum airflow velocity and the airflow velocity in the steady flow zone ($V_{st}$) can be determined using the air volume ($V_{max}$ = 21 m/s, $V_{st}$ = 2.5 m/s, with $Q_{air}$ = 1600 m³/min), independent of the fire location. When $X_L$ is large, a low-velocity zone is formed between the ventilation duct and the fire source as the airflow upstream of the fire source is hindered by the fire plume. At $X_L$ = 50.0 m, the boundary of the turbulent flow transition zone in the upstream direction of the fire source extended to the fire source.

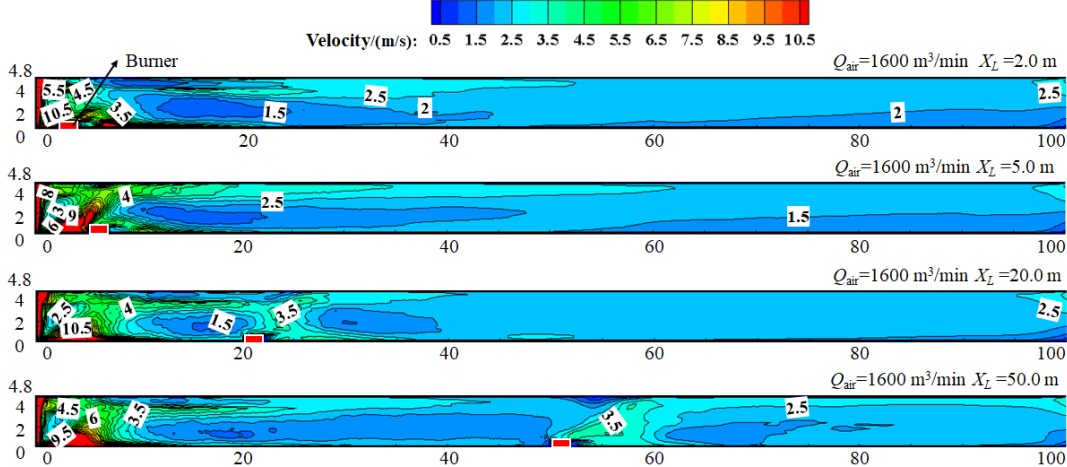

**Figure 7.** Longitudinal flow field distribution of the single-end tunnel under different fire locations.

### 3.2.2. Fire Location Effect on the Temperature Distribution

Figure 8 shows a comparison of the ceiling temperature rise with different fire locations for the case of $Q_{air}$ = 1600 m³/min. The fire location has a significant effect on the longitudinal ceiling temperature rise and the maximum ceiling temperature rise for the cases with the same air volume. The maximum ceiling temperature rise tends to occur downstream of the fire source under the influence of ceiling and ventilation. For the case of $Q_{air}$ = 1600 m³/min, the maximum ceiling temperature rise first decreases from 48.7 °C ($X_L$ = 2.0 m) to 40.8 °C ($X_L$ = 5.0 m), and then gradually increases to 74.3 °C ($X_L$ = 50 m) with the increase in $X_L$. For the case with $X_L$ = 2.0 m, the airflow is restricted by the side wall, followed by a vortex near the excavation face, resulting in upward airflow in the downstream direction. This in return accelerates the upward movement of smoke, leading to a higher ceiling temperature (48.7 °C). For the case with $X_L$ = 5.0 m, the airflow is nearly steady with a higher airflow velocity, resulting in a lower temperature (40.8 °C). As $X_L$ further increases, the influence of the fire source becomes weaker with a decrease in the airflow velocity, with an increase in the maximum ceiling temperature rise. In addition, the fire source affects the excavation face temperature rise, too. The excavation face ceiling temperature rise decreases as $X_L$ increases ($X_L$ < 20.0 m) while remaining stable and small when $X_L$ > 20.0 m. This can be attributed to the fact that the airflow restricting the smoke moves to the excavation face.

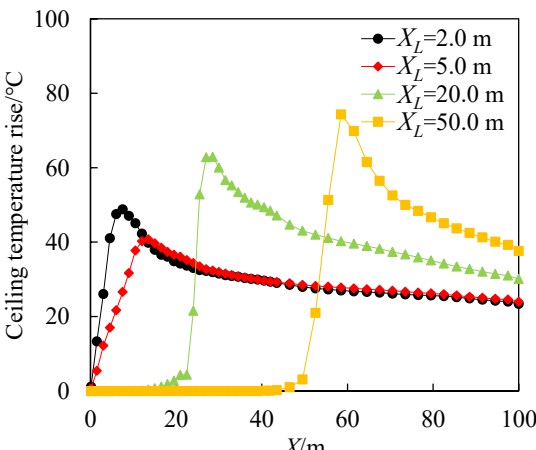

**Figure 8.** Longitudinal ceiling temperature rise under different fire locations.

The detailed temperature distribution from the average over the steady period for the case of $Q_{air}$ = 1600 m³/min is shown in Figure 9. It can be seen that the temperature distribution of the fire source downstream is similar. However, the high-temperature regions (larger than 60 °C) varies with the fire source location, which can be attributed to the inclination of the fire plume [18]. For the case with $X_L$ = 2.0 m, the fire source is close to the excavation face, and the effect of the airflow on the smoke movement is small, as shown in Figure 5. As a result, the temperature of the excavation face can reach above 40 °C, in which the body's vital center will be threatened, and the internal circulatory system will be disordered [25–27]. The upstream temperature near the fire source decreases as $X_L$ increases, which implies that the effect on trapped people would gradually decrease. The downstream temperature rise increases as the fire source gradually approaches the tunnel exit. As the fire location changes from $X_L$ = 2.0 m to $X_L$ = 50.0 m, the height of the 40 °C isotherm decreases from 2.0 m to 0.6 m, which is related to the smoke sinking effect that strengthens the mixing of smoke and fresh air.

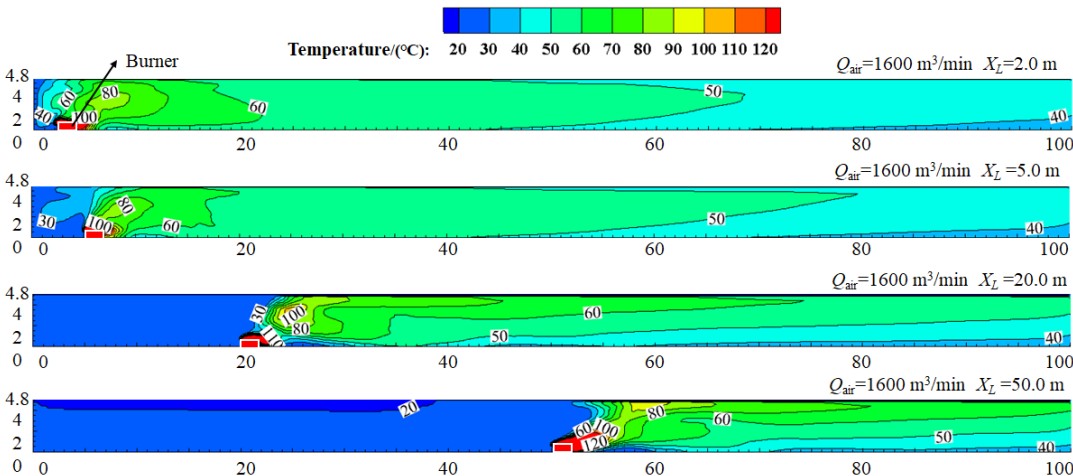

**Figure 9.** Longitudinal temperature distribution of the single-end tunnel under different fire locations.

### 3.2.3. Fire Location Effect on the CO Concentration Distribution

CO concentration is an important parameter related to human safety in the case of a tunnel fire [28]. Figure 10 shows the CO concentration in the tunnel at the height Z = 1.6 m (corresponding to the human breathing zone), averaged over the steady period for the cases with different fire locations. To better understand the distribution of the CO concentration, the flow field distribution at the same height is plotted in Figure 11.

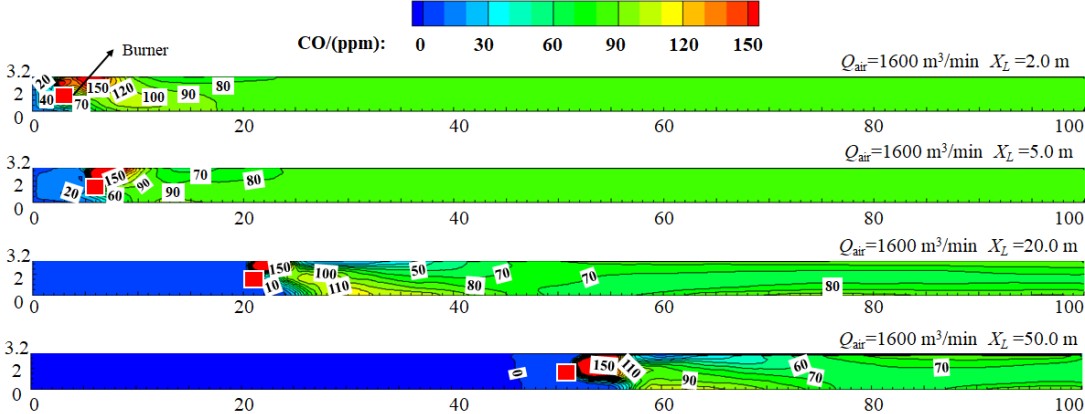

**Figure 10.** The CO concentration distribution at the height Z = 1.6 m under different fire locations.

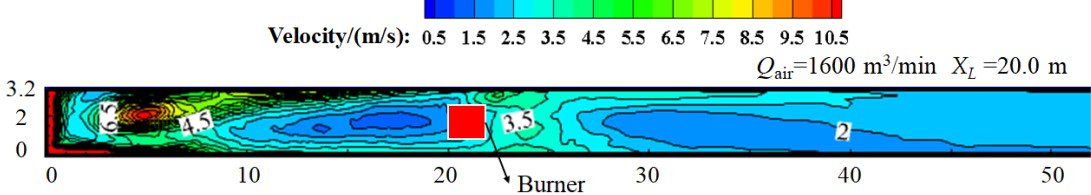

**Figure 11.** The flow field distribution at the height Z = 1.6 m. ($Q_{air}$ = 1600 m$^3$/min, $X_L$ = 20.0 m).

It can be seen in Figure 10 that the CO concentration in the downstream direction is generally higher than that in the upstream direction for a given air volume. This is because the smoke moves downstream by forced ventilation, resulting in CO moving towards the tunnel exit. The CO concentration near the fire source is highest indicating the combustion region. Near the fire source, the CO concentration on the side of the ventilation duct is higher than that on the other side, which can be attributed to the fact that the ventilation duct limits the upward diffusion of smoke, so smoke can easily accumulate under the

ventilation duct. However, in the zone (5.0–15.0 m downstream of the fire source), the CO concentration on the opposite side of the ventilation duct is higher than that on the ventilation duct side. This is because of the existence of the small airflow velocity area, as shown in Figure 11. Based on the above analysis, it can be noted that the threat to trapped people increases as $X_L$ decreases for a given air volume. Special attention should be paid to the two areas (the side of the ventilation duct near the fire source and the opposite side of the ventilation duct 5–15 m downstream of the fire source), in which CO can easily accumulate, which can impact on the evacuation and rescue in real single-ended tunnel fire accidents.

### 3.3. Forced Air Volume Effect on the Tunnel Environment

It has been shown that the length of the turbulent flow transition zone depends on the air volume. To further examine the influence of air volume on the internal environment of the tunnel, simulations were performed for the case with $X_L$ = 5.0 m and the air volume set to 400, 800, 1200, and 1600 m$^3$/min.

### 3.3.1. Forced Air Volume Effect on the Flow Field Distribution

Figure 12 shows the distribution of the flow field in the longitudinal direction, averaged over the steady period for the cases with different air volumes. The airflow velocity near the excavation face increases with increasing air volume. This is because the momentum of the airflow at the ventilation duct mouth increases as the air volume increases. Meanwhile, the flow field downstream of the fire source gradually becomes uniform as the air volume increases. For the case with $Q_{air}$ = 400 m$^3$/min, there is a significant stratification of the flow field in the downstream direction of the fire source, whereas for the case with $Q_{air}$ = 1600 m$^3$/min, the velocity stratification gradually disappears as the distance from the tunnel exit decreases.

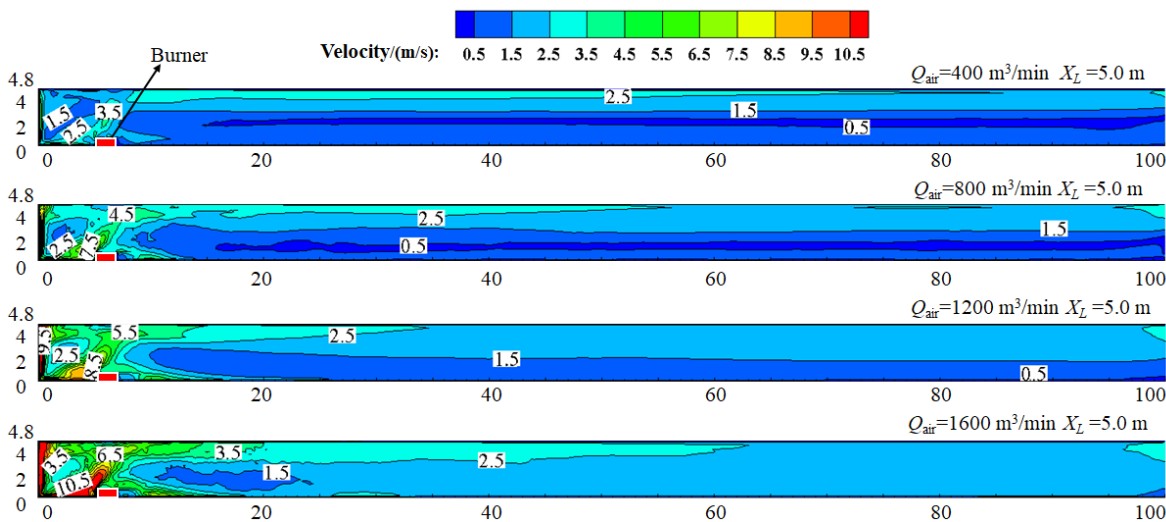

**Figure 12.** Longitudinal flow field distribution of the single-end tunnel under different air volumes.

### 3.3.2. Forced Air Volume Effect on the Temperature Distribution

Figure 13 shows the ceiling temperature rise in the longitudinal direction. The air volume has a great influence on the temperature distribution for a given fire source position, and the larger the airflow velocity, the lower the ceiling temperature rise. This can be attributed to the increased cooling effect with the airflow velocity. The maximum ceiling temperature rise decreases from 134.3 to 41.1 °C, and the tunnel exit ceiling temperature rise decreases from 43.16 to 23.76 °C as $Q_{air}$ increases from 400 to 1600 m$^3$/min. This is mainly because the large airflow velocity results from the fire plume inclination. The ceiling temperature rise decreases as the air volume increases, and the decreasing trend gradually

becomes slow near the excavation face. In practical accidents, it is important to increase the air volume to control the temperature near the excavation face.

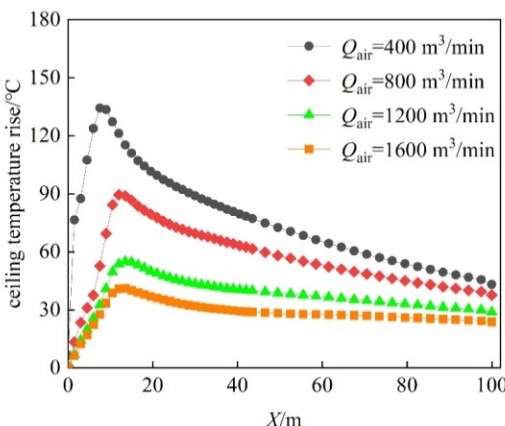

**Figure 13.** Longitudinal ceiling temperature rise under different air volumes.

Figure 14 shows the temperature distribution in the tunnel, averaged over the steady period for the cases with different air volumes. The smoke layers with different temperature gradients are formed in the vertical section. The temperature near the excavation face increases as the air volume decreases. Taking the cases with $X_L$ = 5.0 m as an example, the temperature at the excavation face decreases from 80 to 30 °C as the air volume increases from 400 to 1600 m$^3$/min. However, the area of high-temperature regions (larger than 40 °C) near the ground increases as the air volume increases, and the smoke stratification is not obvious. This would increase the evacuation and rescue risk in the downstream direction of the fire.

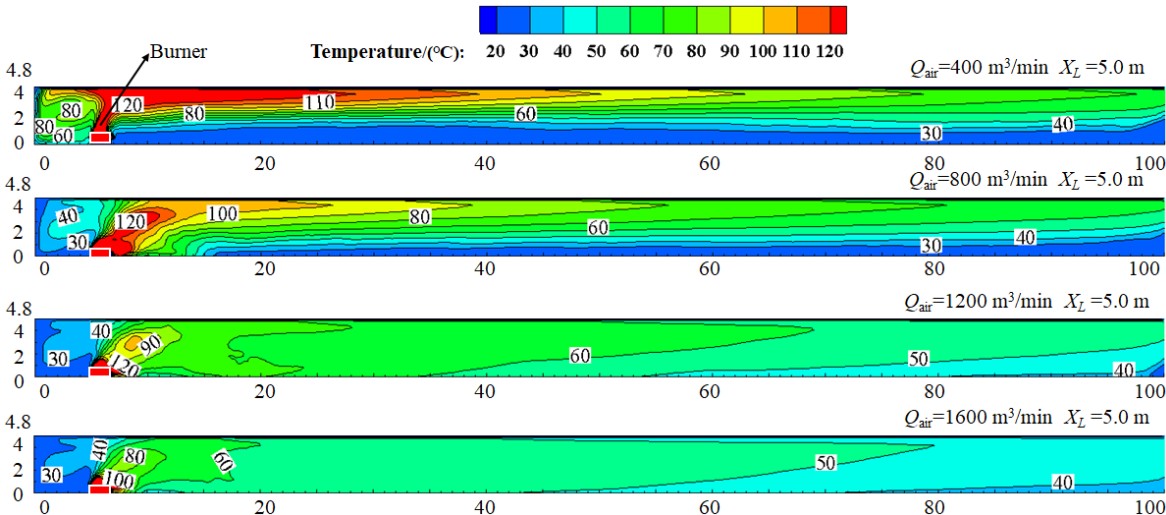

**Figure 14.** Longitudinal temperature distribution of the single-end tunnel under different air volumes.

### 3.3.3. Forced Air Volume Effect on the CO Concentration Distribution

Figure 15 shows the CO concentration in the tunnel at the height Z = 1.6 m, averaged over the steady period for the cases with different air volumes. The CO concentration in the excavation face decreases with the increase in air volume. For example, it decreases from 110 to 10 ppm as the air volume increases from 400 to 1600 m$^3$/min. This is because more smoke moves to the downstream area with the increase in air volume. In the downstream direction of the fire source, the CO concentration first increases as the air volume increases, and then decreases when the air volume exceeds a certain value. During the initial increases, the increased air volume promotes the convection of the smoke layer and the air layer. For large

airflow ($Q_{air}$ = 1200 m$^3$/min), the smoke occupies the whole tunnel in the downstream direction and the distribution of CO concentration is almost uniform. After the initial mixing, the CO concentration in the downstream direction of the fire source decreases due to dilution [29]. A relatively high CO concentration area can be found near the tunnel exit when the air volume is small. For example, for the case with $Q_{air}$ = 400 m$^3$/min, the CO concentration is higher at $X$ = 70.0–90.0 m. This is because of the sinking of the smoke as the smoke temperature decreases [30].

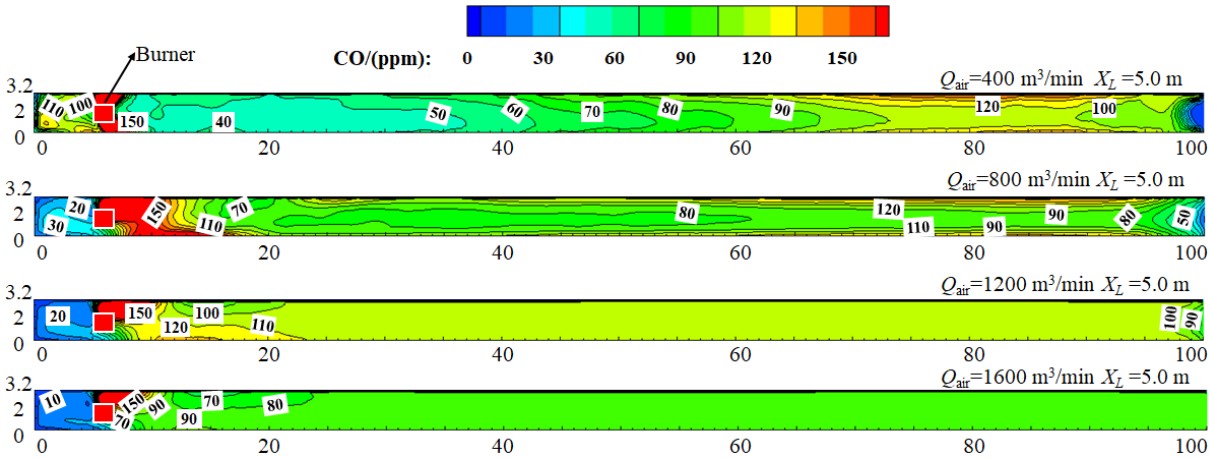

**Figure 15.** The CO concentration distribution at the height Z = 1.6 m under different air volumes.

## 4. Conclusions

In this study, the effects of air volume and fire location on the environment of a single-ended tunnel with a forcing system were investigated using CFD. The flow field distribution, temperature distribution, and CO concentration distribution were analyzed. The main findings are as follows:

(1) In the single-ended tunnel with forced ventilation, the flow field can be divided into three zones according to the direction and velocity of airflow: (1) turbulent flow zone, (2) turbulent flow transition zone, and (3) steady flow zone. For the turbulent flow zone, the airflow is chaotic, and a vortex zone is formed between the ventilation duct and the excavation face. For the turbulent transition zone, the airflow direction remains stable in the longitudinal direction, while the airflow velocity is not uniform with the velocity higher near the ceiling position. For the steady flow zone, both the flow velocity and the direction are stable in the tunnel.

(2) In the single-ended tunnel fire accident, the maximum temperature at the excavation face is sensitive to the fire location. The maximum ceiling temperature rise decreases first with an increase in the distance to the excavation face ($X_L$), and then increases as $X_L$ further increases, because of the presence of the airflow vortex. The CO concentration can easily accumulate on the ventilation duct side at the fire source position and the opposite side of the ventilation duct 5.0–15.0 m downstream of the fire source. Therefore, one should avoid staying in these areas during personnel evacuation and firefighting.

(3) Large air volume can promote a safe environment near the excavation face in a fire accident. Both the CO concentration and the maximum ceiling temperature rise decrease obviously with increasing air volume, which is beneficial for trapped people. However, the environment in the downstream direction of the fire source will worsen to some extent for the evacuation of personnel due to the mixing effect under strong convection. This phenomenon becomes more profound after a critical air volume value.

These findings highlight the importance of comprehensively considering the coupling effect of fire location and forced air volume when a fire accident occurs in a single-ended tunnel with forced ventilation and adjusting the air volume of the ventilation duct in time to ensure the safety of trapped people and the safe evacuation of personnel.

**Author Contributions:** Conceptualization, J.Z. (Jinlong Zhao) and Z.W.; methodology, J.Z. (Jinlong Zhao); validation, Z.W. and J.Z. (Jianping Zhang); formal analysis, J.Z. (Jianping Zhang); investigation, X.P. and Z.H.; resources, J.Z. and Z.W.; writing—original draft preparation, J.Z. and Z.W.; writing—review and editing, Z.W., J.Z., and X.C.; visualization, X.P.; supervision, X.P. and J.Z. (Jianping Zhang); project administration, X.P.; funding acquisition, J.Z. (Jinlong Zhao). All authors have read and agreed to the published version of the manuscript.

**Funding:** This research was funded by the Fire and Rescue Department Ministry of Emergency Management (No. 2022XFZD04), the National Natural Science Foundation of China (No. 51906253), and the Fundamental Research Funds for the Central Universities (No. 2020QN05).

**Institutional Review Board Statement:** Not applicable.

**Informed Consent Statement:** Not applicable.

**Data Availability Statement:** The authors declare that the data supporting the findings of this study are available within the article.

**Conflicts of Interest:** The authors declare no conflict of interest.

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
