# Peer review of "Effects of Fire Location and Forced Air Volume on Fire Development for Single-Ended Tunnel Fire with Forced Ventilation"

_fire, doi:10.3390/fire6030111_

Round 1
Reviewer 1 Report
The fire location and air volume effects on smoke distribution in single-ended tunnel fire with forced ventilation were studied systematically. The study has good academic significance to the trapped persons in the tunnel when fire accidents occur. Generally speaking, this work is carefully drafted and has meaningful application in the rescue process. I recommend its publication, however, some improvements need to be made.
1. The phenomenon of mixing effect at fire source downstream will affect the person evacuation and eventually determine the forced wind strategy. This mixing effect is very interesting. Please have a description on how to reduce this mixing effect by ventilation strategy.
2. It is suggested to divide the second paragraph of Introduction into several paragraphs in order to increase the readability of the article.
3. Page 4, line 138, please cite the reference in this sentence “The CO yield of the fire-resistant belt was set as 0.06, based on the test data obtained in cone calorimeter tests.”.
4. Some format improvements, such as Page 2, line 64, Chen et al. [4][5] uses [4-5]; Page 3, line 115, Table 1 [16][17] uses [16-17]; Page 4, line 136, ‘tunnel, In the simulation’ should have a period in the middle instead of commas; Page 12, line 357, Figure.15 should be Figure 15...Please check the full text carefully.
Reviewer 2 Report
The authors paid more attention to examine the effects of fire location and air volume on smoke in single-ended tunnel fire with forced ventilation by FDS method. In the study, the flow field, temperature, and CO concentration distribution inside the tunnel were measured and analyzed. This topic is interesting and meaningful. The manuscript is well structured.
There are some minor problems should be addressed before publication.
(1) The abstract is too long. The detail information such as Han et al. should be deleted.
(2) Some explanations for the experimental phenomenon should be enriched.
(3) Some words in the pictures are unclear.
(4) There are some grammar mistakes that should be corrected.
Reviewer 3 Report
Effects of fire location and forced air volume on the fire development for single-ended tunnel fire with forced ventilation.
The proposed paper is an interesting approach for tunnel safety during its construction period.
Of course, it would be perfect to combine actual real scale experiments and simulations. In case of given literature, the paper is lacking in discussion about achieved results from the simulations and the results from empirical experiments.
Also, people safety could be described more detailed. Relying only on CO parameter is somehow meager comparing to the possibilities given by the FDS software. Also the fire curve and other specifications are not described well enough.
Detailed remarks to lines:
21 – Please add any clarification to FDS acronym (full name or just FDS simulation software)…
39 – Please consider any explanation about different types of tunneling. There might be difference between TBM techniques and mining methods.
109 – Fire Dynamics Simulator should start with capital letters. It would be perfect to describe the developer of the software.
137 – The fire development is not described well enough. Please add alfa parameter. Please add soot yield. Please add heat of combustion. Is the combustion model enabled? What was the achieved fire curve after the simulations?
286 – Please consider adding as a parameter related to fire safety odder factors: smoke concentration/visibility, oxygen concentration, other toxic gases… CO is produced when the fire is mostly controlled by ventilation. Was that considered?
